# Bibliometric Analysis of the Scientific Productivity on Functional Properties and Enzymatic Hydrolysis of Proteins from By-Products

**DOI:** 10.3390/foods14213693

**Published:** 2025-10-29

**Authors:** Sebastián Plaza, Suleivys M. Nuñez, Yunesky Masip, Pedro Valencia

**Affiliations:** 1Programa de Magister en Ingeniería de Procesos, Escuela de Ingeniería Química, Pontificia Universidad Católica de Valparaíso, Valparaíso 2340025, Chile; sebastian.plaza.d@mail.pucv.cl; 2Escuela de Ingeniería Química, Pontificia Universidad Católica de Valparaíso, Valparaíso 2340025, Chile; 3Escuela de Ingeniería Mecánica, Pontificia Universidad Católica de Valparaíso, Valparaíso 2340025, Chile; yunesky.masip@pucv.cl; 4Escuela de Alimentos, Pontificia Universidad Católica de Valparaíso, Valparaíso 2340025, Chile; pedro.valencia@pucv.cl

**Keywords:** protein hydrolysates, functional properties, bioactive peptides, enzymatic hydrolysis, bibliometric analysis, VOSviewer, biblioshiny, valorization

## Abstract

The growing interest in functional foods has driven research into protein hydrolysates produced by enzymatic hydrolysis, especially from agro-industrial by-products. These compounds stand out for their antioxidant, antihypertensive, and other bioactive properties, which are relevant to the food, pharmaceutical, and nutraceutical industries. In this context, a bibliometric analysis of 1498 articles indexed in the Web of Science (WoS) database (2015–2025, collected up to June) was conducted to map the evolution of knowledge, identify consolidated and emerging thematic lines, and recognize the most influential actors in the field. The methodology combined an advanced search strategy with Biblioshiny (RStudio) and VOSviewer to generate co-occurrence maps, collaboration networks, and citation analyses. The results show sustained growth since 2018, with a predominance of research on functional properties, bioactive peptides, and antioxidant activity, along with a growing interest in sustainability, process optimization, and in silico methodologies. Six thematic clusters were identified, encompassing process optimization, biofunctional validation, circular economy, and non-conventional protein sources. The study concludes that the field demonstrates significant thematic maturity, with opportunities for innovation, particularly in functional validation and the integrated use of by-products through sustainable enzymatic technologies.

## 1. Introduction

The remarkable rise in the demand for functional foods and bioactive compounds has fostered intense research activity on protein hydrolysates, due to their notable functional properties such as antioxidant and antimicrobial activity, angiotensin-converting enzyme (ACE) inhibitory capacity, and improvements in water-holding capacity and solubility, which position them as strategic ingredients for the food, pharmaceutical, and nutraceutical industries [1,2,3,4]. Beyond cataloguing bioactivities, a process-level understanding of how protein hydrolysates are generated is essential because the technological route used to produce them strongly determines their composition and, therefore, their functional performance.

These hydrolysates are mainly obtained through enzymatic hydrolysis. This more controlled, selective, and environmentally friendly process enables the release of bioactive peptides from intact proteins, producing products with specific functionalities without compromising their safety or industrial applicability [4]. In practical terms, enzyme specificity (endo-/exoproteases), enzyme-to-substrate ratio, pH, temperature, and reaction time govern the degree of hydrolysis (DH), peptide length distributions, and sequence motifs—key determinants of solubility, emulsification, antioxidant capacity, ACE inhibition, and antimicrobial effects [5]. Process intensification options (e.g., ultrasound, high-pressure, and microwave processing) and downstream fractionation (membranes, chromatography) further shape yield, purity, and scalability. Thus, there is a clear need to study the enzymatic-hydrolysis technology itself, not only its outcomes, to explain and predict functional properties [6].

The diversity of protein sources encompasses those of animal origin (dairy, collagen) [6], plant origin (soy, legumes) [4,7], and marine origin (fishery products, by-products, algae) [7]. In addition, non-conventional sources (insects, microalgae) [8] further expand the range of raw materials for designing efficient enzymatic processes, characterizing the resulting peptides, and evaluating their biological effects [1,9]. This heterogeneity in substrates, together with varied enzymes and operating conditions, produces a broad spectrum of peptide profiles whose structure–function relationships are not yet fully systematized.

In this context, the circular bioeconomy has gained particular relevance as a driver for the valorization of protein-rich agro-industrial by-products, transforming them into high-value-added ingredients [10]. In countries such as Chile, where the fishing and agri-food industries generate large volumes of organic waste annually, utilizing these by-products through enzymatic hydrolysis represents a strategic opportunity from both environmental and economic perspectives [11,12]. However, realizing this opportunity requires robust evidence linking process descriptors (e.g., DH, enzyme type, fractionation route) with functional performance in targeted applications.

Nevertheless, despite sustained growth in the field, the scientific literature shows thematic dispersion, hindering the identification of consolidated research lines, emerging topics, and key actors in its development [13]. Previous bibliometric studies have addressed specific areas, such as antioxidant and antidiabetic peptides, as well as trends in functional marine peptides [14]. Still, there is no integrative analysis that simultaneously encompasses enzymatic protein hydrolysis, the functional and bioactive properties of its derivatives, and their relationship with by-product valorization [15]. Given the methodological diversity and the multi-dimensional nature of outcomes, narrative reviews are insufficient to capture consistent patterns. A specialized statistical analysis of the literature is therefore required to quantify method–outcome linkages and resolve fragmentation across substrates, enzymes, and assays.

In this regard, bibliometrics emerges as an effective tool for exploring the evolution of scientific knowledge, mapping collaboration networks, detecting high-impact areas, and highlighting opportunities for future research [13,16]. Specifically, science mapping (co-word/co-occurrence), co-citation and bibliographic coupling, thematic evolution, and strategic diagrams (centrality–density) can uncover the technological backbone of enzymatic hydrolysis (enzymes, DH control, fractionation) and its association with functional endpoints. Complementary multivariate techniques (e.g., multiple correspondence analysis or topic modeling) can further quantify these associations.

Therefore, this study is based on the hypothesis that bibliometric analysis provides a suitable and objective methodological framework for identifying relationships between process variables and functional outcomes, given the multidimensional, heterogeneous, and interdisciplinary nature of research on enzymatic hydrolysis and bioactive peptides. By quantitatively mapping scientific production and thematic evolution, bibliometrics provides a justified means to integrate scattered knowledge and reveal the conceptual structure of this research field.

To this end, a corpus of 1498 articles from the Web of Sciencie © (WoS) database (2015–2025, collected up to June) was analyzed, selected through a rigorous search strategy focused on terms such as “enzymatic hydrolysis of proteins”, “bioactive peptides”, “functional properties”, “bioactive properties”, “by-products”, and “waste valorization” [17,18], using tools such as Biblioshiny (RStudio) and VOSviewer. A structured keyword screening and harmonization workflow was implemented to minimize bias and ensure reproducibility, and co-word networks, thematic mapping, and coupling analysis were applied to link process descriptors with reported functions. The primary objective is to identify trends, key actors, and knowledge gaps that can inform future research on the development of functional foods, dietary supplements, and waste valorization strategies [19,20,21]. In addition to offering an evidence-based roadmap that aligns enzymatic hydrolysis parameters with functional/bioactive performance to guide scalable, low-impact valorization pathways.

## 2. Materials and Methods

Data collection for this study was conducted using the Web of Science (WoS) database, a globally recognized resource for its rigorous indexing of high-impact scientific literature and for providing standardized metadata that enable high-quality bibliometric analyses [22,23,24]. To accurately capture scientific literature on enzymatic protein hydrolysis, its functional properties, and its relationship with by-product valorization, an advanced search strategy was structured in the “Topic” (TS) field. Boolean operators were used (AND for combining terms, OR for synonyms), with related terms grouped in parentheses, to precisely delimit the thematic scope of interest and minimize irrelevant results.

The complete search sequence used was: TS = (“enzymatic hydrolysis of proteins” OR “protein enzymatic hydrolysis” OR (“functional properties” OR “bioactive properties” OR “bioactive peptides”) AND (“by-products” OR “byproduct” OR “waste valorization”)). It should be clarified that, in the Web of Science (WoS) search interface, each thematic block was entered separately as an independent topic set (TS) and then combined using the logical operator AND to ensure the correct Boolean structure of the final query.

The search was defined between 2015 and 2025 (data extracted up to 5 June 2025). Since 2015, a sustained increase has been observed in scientific productivity related to the functional properties of protein hydrolysates and their production from by-products [10,25,26]. This time frame excludes the early years, in which scientific output was significantly lower, allowing the analysis to focus on the most recent and consolidated trends in the field.

The search was limited to English-language documents and restricted to the document types Article or Review to ensure inclusion of original research and high-level scientific reviews [27,28,29]. In addition, the research areas were delimited through the WC (Web of Science Categories), including Food Science Technology, Chemistry, Biochemistry, Molecular Biology, Engineering, Nutrition Dietetics, Biotechnology Applied Microbiology, Science Technology Other Topics, Pharmacology Pharmacy, and Microbiology. As a result, 1498 documents were retrieved, comprising 1276 original articles and 222 review articles.

To structure the query, three main thematic sets were organized:Technological processes: related to enzymatic protein hydrolysis (“enzymatic hydrolysis of proteins”, “protein enzymatic hydrolysis”).Functional properties: focused on bioactive compounds and protein functionality (“functional properties”, “bioactive properties”, “bioactive peptides”).Raw material and sustainability: focused on by-product valorization (“by-product”, “byproduct”, “waste valorization”).

These sets were intersected using the AND operator to ensure that the selected documents simultaneously addressed the three central thematic axes of the study. To provide a concise, visual representation of the methodological stages, a flowchart was developed to summarize the process of information collection, processing, and analysis, as shown in Figure 1.

The retrieved data were analyzed using Biblioshiny, an interactive graphical interface that operates within the RStudio (v2025.05.1) and R (v4.5.1) statistical programming environment. [30,31]. The environment was executed through the command Bibliometrix: biblioshiny in the R console, allowing access to the full functionality of the Bibliometrix package for bibliographic mining, productivity analysis, impact assessment, scientific collaboration, and terms “by-product” and “by-products” appeared as variants due to the way they were used by the authors in the original articles. In the present study, both forms were considered equivalent and grouped under a single descriptor (by-products) to avoid duplication and ensure consistent interpretation of the results.

During data processing, cleaning procedures were applied to remove duplicate records, correct inconsistencies in author and institution names, and normalize keyword and metadata terms. Subsequently, a final database consisting of 1498 documents was consolidated and imported into the Biblioshiny environment, a graphical interface of the Bibliometrix (v5.0.1) package in RStudio, for further bibliometric analysis and processing.

At this stage, filters and analytical tools were applied to identify the ten most relevant bibliometric indicators for the study. These include “Main Information”, “Annual Scientific Production”, “Average Citations per Year”, “Most Relevant Sources”, “Most Relevant Authors”, “Most Relevant Affiliations”, “Most Cited Countries”, “Most Global Cited Documents”, “Most Frequent Words”, and “Thematic Map”.

Additionally, complementary visualizations were incorporated, such as the “Three-Field Plot”, “Countries’ Collaboration World Map” journal co-citation analysis, and the keywords co-occurrence network “Co-occurrence Network”. These tools enabled the characterization of collaboration patterns, thematic research cores, and emerging trends, providing a strategic overview of the field’s evolution, maturity, and structure. Figure 1 summarizes the methodological procedure followed, from record retrieval and filtering to data analysis and interpretation, facilitating the identification of global patterns, key actors, and consolidated themes related to the study of protein hydrolysates and their functional properties. Likewise, VOSviewer (v1.6.20) was used to construct maps of relationships among keywords, authors, countries, and institutions, thereby allowing the identification of conceptual cores, emerging research lines, and academic collaboration dynamics [32,33].

## 3. Bibliometric Analysis

The analysis of the 1498 documents retrieved from WoS (2015–2025, collected up to June) reveals a steadily growing scientific output in the field of enzymatic protein hydrolysis from by-products [34,35]. In the Biblioshiny environment, filters were applied to identify the 10 most relevant bibliometric indicators, and the results are presented in Figure 1. Among these, the most notable are annual production, average citations, the most active sources, countries with the highest number of publications, and the most frequently used keywords.

### 3.1. Evolution of Publications and Research Lines

Figure 2 depicts the annual evolution of the total number of scientific publications (red line) and the percentage distribution of the ten main thematic lines (stacked bars) re-lated to enzymatic protein hydrolysis from by-products during the period 2015–2025, collected up to June, highlighting sustained growth since 2018 in research on functional protein hydrolysates, driven by the interest in sustainable waste valorization and the bioactive properties of peptides [36,37]. The apparent decline in 2025 is explained by the fact that the analysis was conducted using data up to June, so many publications had not yet been indexed. This trend reflects the intensification of research aimed at producing functional peptides, in parallel with the consolidation of a sustainable framework for the valorization of agro-industrial waste [38,39].

At the thematic level, the most frequent terms were “functional properties,” “bioactive peptides,” “antioxidant activity,” and “by-products,” with stable proportional variations throughout the decade analyzed. The relative increase in the categories “extraction,” “physicochemical properties,” and “dietary fiber” over the past five years suggests a broadening of focus toward technological, structural, and nutritional functionalization aspects. [40,41]. It is essential to clarify that occurrences of similar terms such as “functional Properties”, “functional-Properties”, “antioxidant activity”, and “antioxidant” result from differences in keyword indexing in the original database. These terms were kept as indexed to preserve the fidelity and accuracy of the bibliometric data.

The graph also reveals a fragmented yet stable thematic structure, with the coexistence of classical lines such as “antioxidant activity” and “bioactive peptides,” alongside more recent ones linked to sustainability, such as “by-products”. The parallel behavior of article volume and the thematic distribution indicates a growing maturity of the field, with a concentration on functional and methodological topics [42,43].

Table 1 presents a consolidated summary of the main focuses of scientific production in the field of enzymatic protein hydrolysis and functional properties, based on the analysis of 1498 articles indexed in (WoS) between 2015 and 2025. Regarding the predominant research lines, the terms “functional properties” (34.85%), “antioxidant activity” (16.69%), and “bioactive peptides” (14.35%) stand out, reflecting a sustained interest in evaluating the bioactivity of peptides derived from food by-products. These properties have been widely studied for their antioxidant, antihypertensive, antimicrobial, and immune-modulating capacities, which are attributed to peptides generated through enzymatic hydrolysis from diverse protein sources [25,26,44,45].

Likewise, an increasing emphasis can be observed in lines such as “physicochemical properties,” “identification,” and “optimization,” which reflects a focus on the precise structural characterization of peptides, the design of functional ingredients, and the adjustment of processing conditions that maximize their yield and bioactivity [25,44].

Among the most productive affiliations, the Spanish National Research Council (CSIC, Spain), Jiangnan University (China), and CONICET (Argentina) stand out, reinforcing the idea of an international distribution of knowledge driven by centers with a substantial investment in food biotechnology. This pattern is consistent with studies that also position China, Brazil, India, and Spain as leaders in the development of by-product valorization technologies [44].

As for the most active publication channels, journals such as “Foods,” “LWT—Food Science and Technology,” and “Food Chemistry” stand out, having maintained a high publication rate in this field. These journals focus on the processes of production, characterization, and functional application of protein hydrolysates, including those obtained through microbial fermentation [45].

Finally, the most prolific authors identified in the analyzed database, such as Wubshet SG, Afseth NK, Li Y, Oliveira MBPP, and Zhang Y, have made significant contributions to the development of analytical methodologies, functional assays, and the design of peptides with specific activities. The influence of the most relevant participants was determined using bibliometric indicators, including the number of publications, total citations, and network centrality, which reflect both productivity and scientific impact within the field. Their sustained and specialized scientific output has consolidated their academic leadership in this expanding field [26,44].

### 3.2. Most Relevant Keywords

Keyword analysis is a crucial tool in bibliometric studies, as it enables the identification of the primary research lines, emerging trends, and thematic relationships within a specific field [46,47]. In this study, a co-occurrence analysis was performed using VOSviewer software based on the author-provided keywords (Author Keywords), which are particularly useful for characterizing the scientific evolution of the field of protein hydrolysates and bioactive properties [16]. The data visualization included three complementary approaches: thematic cluster mapping, term density, and temporal evolution.

The cluster visualization in Figure 3a shows a clear thematic grouping of the field into six differentiated clusters, as described in more detail in Table 2. A distinct color within the keyword co-occurrence network represents each cluster. Cluster 1 (red) is the largest and is centered on the valorization of food by-products and their conversion into functional compounds. This group includes terms such as “waste valorization,” “functional food,” “sustainability,” and “polyphenols,” reflecting a strong focus on the circular economy and the integral utilization of waste [48,49].

Cluster 2 (green) highlights a classical line of research focused on the functional and structural properties of hydrolysates, including “rheology,” “texture,” and “phenolic compounds.” This group represents a solid foundation for the development of food applications with specific technological and functional properties [50].

In contrast, Cluster 3 (blue) combines elements of process optimization and physicochemical characterization, with terms such as “response surface methodology,” “ultrasound,” and “techno-functional properties,” indicating a methodological evolution toward more sophisticated processing and modeling techniques [51].

Cluster 4 (yellow) encompasses terms related to the hydrolysis process (“enzymatic hydrolysis”), raw materials (“whey” and “okara”), and intermediate products (“antioxidant peptides,” “bioactive peptides”), reflecting the biotechnological core of the field of study [52].

Cluster 5 (purple) groups studies focused on molecular and in silico validation, with terms such as “molecular docking,” “peptide,” and “bioactivity” standing out [53]. Finally, Cluster 6 (sky blue) includes a single term, “functional foods,” which was grouped independently due to its high frequency and cross-cutting relevance in the field. Although it is not part of a broader thematic group, its emergence as an autonomous cluster suggests that it acts as an integrating axis among different lines of research on functional properties, food applications, and the development of bioactive ingredients [54].

In this sense, the keyword clusters reveal a consolidated research core on functional properties and bioactive peptides, while also highlighting emerging lines linked to the circular economy and in silico methodologies, reflecting the diversification and expansion of the field.

In Figure 3b, the average-year analysis reveals a clear emerging trend: terms such as “circular economy,” “optimization,” “plant proteins,” and “molecular docking” appear in yellow-green, indicating an increase in their use over the past three years (2022–2024). This confirms a transition toward more sustainable, computational, and applied approaches, both from a technical and functional perspective [55,56]. Furthermore, the quantitative data reinforce this observation. The terms “circular economy” (41 occurrences), “sustainability” (32), and “molecular docking” (17) show a clear concentration in recent years (2022–2024), according to the co-occurrence analysis performed with VOSviewer. This temporal pattern indicates a growing scientific focus on sustainable valorization strategies and in silico validation methodologies, supporting their identification as emerging themes within the field of functional protein hydrolysates. Figure 3c, corresponding to the density map, shows that the most researched terms “functional properties,” “bioactive peptides,” “antioxidant activity,” and “enzymatic hydrolysis” are concentrated along the central axis of the network.

### 3.3. Thematic Map of Research Domains

Figure 4 presents the thematic map generated from the co-occurrence analysis of the 50 most frequent keywords in the analyzed corpus (2015–2025, collected up to June) using the Biblioshiny thematic mapping algorithm. This tool enables the visual representation of themes along two dimensions: degree of internal development (thematic density; *Y*-axis) and degree of centrality or relevance within the field (*X*-axis). The result is a segmentation into four quadrants that identify motor themes, basic themes, niche themes, and emerging or declining themes [57].

In this study, the “Author Keywords” and the “Index Keywords” are analyzed separately. The “Author Keywords” correspond to the keywords provided directly by the authors in their articles. At the same time, the “Index Keywords” are assigned by the databases during indexing and may not coincide with the authors’ keywords. The latter offers a broader and more standardized categorization, allowing the identification of trends and thematic areas that may not be explicitly indicated in the original text. This distinction justifies the inclusion of two independent analyses, as each provides a complementary perspective of the research landscape.

The quadrant related to basic themes (bottom right quadrant) groups terms such as “bioactive peptides,” “antioxidant activity,” “enzymatic hydrolysis,” “functional properties,” “dietary fiber,” and “bioactive compounds.” These themes show high centrality but low density, indicating that they are fundamental to the field of study, although they could benefit from further internal development. They reflect widely accepted lines of research, but still with room for methodological innovations and emerging applications [58].

As for the niche themes quadrant (upper left quadrant), terms such as “antioxidant,” “protein,” “protein hydrolysate,” “by-product,” “circular economy,” and “sustainability” are found. Although well-developed, these themes exhibit low centrality, suggesting that they are specialized fields, possibly applied in specific contexts or utilized by smaller scientific communities. Their solid thematic structure makes them ideal candidates for specific in-depth studies or applications in concrete products [59].

On the other hand, in the emerging or declining themes quadrant (bottom left quadrant), no representative terms were identified. The absence of keywords in this area of the map could be attributed to the frequency threshold established or to thematic dispersion in the analyzed corpus, suggesting that emerging or declining lines within the field of study have not yet been consolidated [60].

Finally, in the motor themes quadrant (upper right quadrant), no representative terms were identified either. This absence could be due to the keyword threshold used or to the current thematic dispersion, suggesting that there is not yet a thematic line that simultaneously combines high density and high centrality [61].

The analysis of the thematic map reveals that research is grounded in solid bases, such as enzymatic hydrolysis and bioactive peptides, but still lacks consolidated motor themes. In parallel, concepts such as sustainability and the circular economy are emerging strongly, offering promising avenues for the field’s future evolution.

Table 3 describes the five clusters, in line with the parameters of Figure 4, showing the number of elements in each cluster and the most representative keywords. In contrast, Figure 5 presents the “Index Keywords,” which provide a broader, database-based categorization of research topics. This distinction is crucial for understanding how research themes are self-reported and classified at the indexing level. The size of the circle represents the frequency of keyword use within each cluster. The connections among the different groups can be observed from the use of each keyword within each cluster.

As mentioned above, Table 3 presents the five thematic clusters identified through keyword co-occurrence analysis conducted in the Biblioshiny environment, utilizing the “Walktrap” clustering algorithm. This classification reveals various relevant aspects that complement the observations shown in Figure 5. Each cluster group identifies terms that co-occur frequently in the literature, enabling visualization of the predominant conceptual cores in the study of protein hydrolysates and their functional properties.

Cluster 1—functional properties

Whose representative term is “functional properties,” groups 17 keywords related to the technological and bio-functional evaluation of hydrolysates, including aspects such as by-product valorization, physicochemical properties, rheology, and their application in functional foods [62]. In Figure 5, this term is linked to others, such as “enzymatic hydrolysis” and “optimization,” reinforcing the importance of ongoing research to further optimize these technologies and their applicability in the development of functional products. This group represents the most consolidated line of research within the field.

Cluster 2—bioactive peptides

Cluster 2, in turn, is centered on “bioactive peptides” and encompasses 13 terms related to enzymatic hydrolysis, peptide generation, their functional characterization, and process intensification methods, such as ultrasonication or model-based optimization. This grouping highlights the predominant biotechnological focus on generating functional ingredients [63].

Cluster 3—antioxidant

In Cluster 3, with “antioxidant” as the main keyword, terms are concentrated around the evaluation of the antioxidant bioactivity of the peptides obtained, including in vitro and in silico methodologies such as “molecular docking” [64]. Its position in the network indicates an area of growing specialization oriented toward functional validation.

Cluster 4—sustainability

In contrast, Cluster 4, centered on “sustainability,” brings together terms related to plant protein use, waste valorization, and protein extraction, suggesting an emerging line linked to sustainability and the circular economy [65].

Cluster 5—protein extraction

Finally, Cluster 5, represented solely by the term “protein extraction,” reflects a thematic line with low connectivity, which could indicate a particular technical specialization or a still incipient topic within the field of study [66]. Although underdeveloped, this cluster holds scientific significance, as it encompasses methodological advances and technological approaches to improve protein recovery and purification from diverse agri-food by-products. The exploration of extraction techniques, including enzymatic, physical, or combined processes, is a critical step to maximize yield and functionality, thereby linking upstream processing with subsequent hydrolysis and functional validation [67].

This thematic segmentation enables the identification not only of the most active areas but also those with high development potential, thereby providing a valuable guide for future research aimed at the integral utilization of agri-food by-products through the use of sustainable technologies [68].

Table 3 and Figure 5 provide a comprehensive view that confirms that functional properties dominate current research, particularly in antioxidant activity. On the other hand, areas such as sustainability and protein extraction offer opportunities for future research to optimize the use of underutilized protein sources and to contribute to sustainability by reducing organic waste. This structure reinforces the coherence observed in the thematic map by showing how key concepts are interrelated and identifying the dominant cores in scientific production on functional protein hydrolysates. Building on this basis, Figure 5 provides a deeper visualization of the semantic relationships among key terms through a co-occurrence map constructed using VOSviewer. In this graph, the size of the labels indicates the frequency of each term, while the colors differentiate the thematic clusters identified by the Walktrap algorithm.

The five observed groups allow for a strategic interpretation of the field. The red cluster, led by “functional properties” and “by-product” (Figure 5), represents the most consolidated core, linking the functional characterization of ingredients with strategies for valorizing agri-food by-products [17,21]. The purple cluster, centered on “enzymatic hydrolysis,” “bioactive peptides,” and “antioxidant activity,” reflects a dominant methodological axis related to the production of functional peptides and their experimental validation [69,70].

The green cluster, dominated by terms such as “bioactivity,” “antioxidant,” and “peptides,” shows a strong orientation toward investigating specific biological effects using in vitro techniques, highlighting interest in molecular-level functionality [20]. In turn, the blue cluster groups terms such as “sustainability,” “plant protein,” and “food waste valorization,” suggesting a convergence between sustainability, plant protein recovery, and circular economy strategies [71]. Finally, the orange cluster, centered on “protein extraction,” contains fewer connected keywords but may represent emerging topics or cross-cutting technical lines [72]. Taken together, this thematic segmentation enables the identification of critical intersections among functional properties, processing technologies, and sustainability, providing a solid conceptual basis to guide future research and innovation in the field of protein hydrolysates [73].

Table 4 presents the twenty most representative keywords of scientific production in the field of functional protein hydrolysates, ordered by their frequency of occurrence in the analyzed articles and accompanied by their respective connection strength within the semantic network (Total Bond Strength), calculated using the betweenness centrality metric. This combination enables the identification of both the popularity of the terms and their structural role in the thematic articulation of the field [74].

Leading the pack is the term “functional properties,” with 151 occurrences and the highest connectivity value (271), indicating its position as a cross-cutting axis in current studies. It is followed by “bioactive peptides” (104 occurrences) and “antioxidant activity” (84), terms that reflect the consolidated interest in evaluating the biological functionality of peptides obtained through enzymatic hydrolysis [75].

The presence of concepts such as “by-product” (79 occurrences, 240 link strength) and “enzymatic hydrolysis” (72 occurrences) reinforces the field’s orientation toward the valorization of agri-food residues through controlled technological processes [76]. Terms such as “antioxidant,” “dietary fiber,” “bioactive compounds,” and “phenolic compounds” also appear frequently, associated with specific functional properties that guide the valorization of “by-products” [77].

On the other hand, keywords with lower frequency but high relative centrality, such as “by-products” and “protein,” were identified, suggesting their role as connectors between different thematic lines. Likewise, the inclusion of technical terms such as “response surface methodology,” “extrusion,” and “physicochemical properties,” although with low occurrence and semantic density, indicates an interest in optimization methodologies and technological characterization [78]. This table highlights a thematic structure centered on the functionality, sustainability, and applicability of protein hydrolysates, integrating bioactive, technological, and circular economy approaches in the design of high-value-added functional ingredients.

### 3.4. Most Cited Articles Within the Research Theme

Table 5 summarizes the 10 most cited articles in the analyzed corpus (2015–2025, collected up to June), which represent key contributions that have significantly influenced the development of the fields of protein hydrolysates, bioactive peptides, and by-product valorization. The most cited article is [17], with 409 citations and an annual average of 40.9, indicating a high and sustained impact. This work aligns directly with the predominant themes of this analysis, particularly in the use of marine by-products as a source of antioxidant peptides, thereby strengthening the sustainability and functionality axis identified in the thematic clusters [79].

It is followed by article [18], with 348 citations, which, although focused on biomedical applications, illustrates the growing interest in protein-based materials with functional properties applicable to the food and nutraceutical fields [82]. Similarly, article [19], with 277 citations, positions dairy waste as a strategic resource for obtaining bioactive proteins [83], an aspect repeatedly addressed in research lines on by-products and the circular economy.

Particularly noteworthy is article [21], with an average of 52.2 annual citations, the highest in the set. This result reflects the contemporary rise in comprehensive valorization approaches for fishery residues, consistent with emerging terms such as “waste valorization” and “circular economy” [84], located in the lower-left quadrant of the thematic map (Figure 4). This work reinforces the current relevance of the sustainable approach in protein hydrolysate research.

Other relevant articles include [20], which explores the functional potential of plant proteins such as industrial hemp, highlighting the growth of emerging lines associated with “plant proteins,” “optimization,” and plant bio-functionality. Along the same lines, [71] analyzes the use of hemp across various industries, contributing to interest in its potential as a functional biomass. These topics are also reflected in the evolution of keywords, as shown in the temporal analysis in Figure 3b.

Regarding the experimental validation of antioxidant activity, article [70], with 213 citations, provides a representative model for the structural characterization of peptides obtained through enzymatic hydrolysis. Likewise, a study [69], with 241 citations, reinforces the importance of whey as a matrix for generating functional peptides through controlled processes. In contrast, [80], with 203 citations, examines the multisectoral potential of agri-food residues and highlights their contribution to the circular economy.

Finally, article [81], with 181 citations, is fully aligned with the three central thematic axes of the present analysis: enzymatic hydrolysis, functionality, and by-products, consolidating its position as a paradigmatic reference within the field.

As for the origin of the publications, high-impact scientific journals such as Journal of Functional Foods, Biotechnology Advances, and Marine Drugs stand out, which are also among the most active, as ranked in Table 1. This pattern confirms the link between scientific impact (citations) and editorial productivity (number of articles), particularly in journals that address topics such as bioactivity, biocatalysis, and functional applications. These aspects are also reflected in the evolution of keywords, as shown in the temporal analysis (Figure 3b), where concepts such as “protein hydrolysate,” “antioxidant activity,” and “waste valorization” stand out [85].

The most cited articles confirm that the valorization of marine, dairy, and plant by-products, along with the validation of functional activity, are the axes that have had the most significant impact over the last decade, reinforcing the link between sustainability and bioactivity as drivers of research.

### 3.5. Scientific Productivity by Authors, Journals, Institutions, and Countries

Scientific production in a given field can be analyzed from multiple dimensions, with authorship being one key indicator for identifying the most influential actors [86]. In the present study, the individual productivity of the most prominent researchers in the field of functional protein hydrolysates was evaluated, considering both the total number of publications and their temporal distribution during 2015–2025 [87].

Figure 6 illustrates the evolution of the scientific output of the most prolific authors in the field, presented through a temporal scatter plot. The size of the circles indicates the number of articles published by each author in a given year. At the same time, the color intensity represents the average number of citations per year (TCpY), allowing for the simultaneous visualization of both productivity and academic impact [88].

Noteworthy is Wubshet SG, whose research activity has been sustained throughout the decade, resulting in 5 publications and a medium-high impact [89]. He is followed by Afseth NK, with a comparable number of publications, characterized by a continuous trajectory and a balanced temporal distribution. Meanwhile, authors such as Li Y, Oliveira MBPP, and Zhang Y show peaks of scientific activity concentrated in specific years (2020–2023), which could reflect the consolidation of particular research lines or participation in short-term collaborative projects with high visibility [90]. It is worth noting that the color intensity of some nodes, such as those corresponding to Wu JP or Ma HL in 2021, indicates a higher annual average of citations, suggesting that specific articles have had a particularly significant impact, even if they do not represent the largest number of publications [91]. This phenomenon is related to the concept of “productivity relative to impact,” according to which a lower volume of publications can translate into greater visibility and scientific influence when emerging topics are addressed or innovative methodologies are applied [92].

Upon reviewing the data on average citations per year (TCpY), it was observed that recent articles by authors such as Wubshet SG (TCpY = 4.25 in 2022) and publications in journals such as Food Chemistry and Food Control have achieved relatively high impact in a short time. This suggests rapid acceptance within the scientific community, especially when addressing analytical methodologies for the functional validation of bioactive peptides [93]. Therefore, the temporal analysis shows a maturation of the field, with several authors maintaining a constant presence throughout the period and others emerging recently with high-impact publications [94].

This trend suggests a combination of already consolidated research cores and the incorporation of new scientific actors, especially since 2020, coinciding with the rise in studies focused on sustainability and by-product valorization [95]. This result, as shown in Table 1 regarding the leading authors, reinforces the evidence of their central role in the field [96]. This type of visualization, by integrating temporal production with citation metrics, makes it possible not only to identify the most active researchers but also those with the most significant influence on the thematic development of the field, thus providing a strategic tool for establishing scientific collaboration networks, identifying opinion leaders, and focusing future searches for specialized literature [97]. Therefore, scientific productivity in this field is concentrated in a small core of authors and reference institutions, evidencing academic leadership poles that guide the global development of the area.

### 3.6. Co-Citation Analysis of Scientific Journals

Figure 7 presents the co-citation analysis among scientific journals based on the 50 most cited sources in the bibliographic corpus for the period 2015–2025, collected up to June. This map, generated using VOSviewer under the co-citation analysis approach, enables the identification of thematic groupings of journals that tend to be cited together, revealing affinities in terms of methodological approaches, technological applications, or shared disciplinary areas [98].

Three main clusters were identified (red cluster, green cluster, and blue cluster). The red cluster comprises generalist and multidisciplinary journals in food science and technology, including Food Chemistry, Journal of Agricultural and Food Chemistry, Molecules, Nutrients, and Journal of Functional Foods. This group reflects the central axis of research focused on bioactive properties, technological functionality, and in vitro studies of peptides obtained from by-products [99]. The green cluster encompasses publications focused on food processing and technology, including LWT—Food Science and Technology, the Journal of Food Engineering, the Journal of Food Science, Meat Science, and the International Journal of Food Science & Technology, among others. These publications address aspects such as process design, functional ingredient engineering, physicochemical characterization, and technological validation, highlighting the link between enzymatic hydrolysis and the development of functional food products [100]. Finally, the blue cluster encompasses journals related to biomedical and applied chemistry, including Food Hydrocolloids, International Journal of Biological Macromolecules, Carbohydrate Polymers, Ultrasonics Sonochemistry, and Innovative Food Science & Emerging Technologies, among others. These fields are intensely focused on the structural characterization of macromolecules, new extraction techniques, process intensification, and advanced physicochemical studies. This group reflects the convergence between process engineering and protein chemistry, indicating an evolution toward emerging analytical methodologies and processes for obtaining high-value-added hydrolysates [101].

The co-citation structure observed reveals a precise and complementary thematic segmentation, in which studies with greater reliance on biochemical, technological, and engineering approaches tend to consolidate into specific but interconnected clusters. This suggests the existence of an interdisciplinary field that integrates food biotechnology, process engineering, and nutritional science to address by-product valorization through the production of functional hydrolysates [102]. This co-citation pattern confirms that research on protein hydrolysates is built from specialized but interdependent communities, reflecting the interdisciplinary and complementary nature of the field.

### 3.7. International Collaboration

Figure 8 combines two complementary visualizations to analyze the geographical landscape of scientific production on protein hydrolysates and their associated functional properties. Figure 8a displays a map of international scientific collaboration, illustrating co-authorship networks among countries and highlighting the most frequent relationships based on the number of shared publications. Figure 8b presents a world map of scientific productivity, where the color of each country reflects the total number of articles published from 2015 to 2025, as of June. Both figures enable the identification of the main geographical poles of contribution to the field, as well as the cooperation structures that drive knowledge generation on an international scale [103].

Figure 8a shows a co-authorship network among countries, characterized by well-defined clusters and a high density of interactions. China stands out as the central node in the network, not only for its volume of publications but also for the intensity of its links with multiple countries. The United States, India, Spain, and Italy also occupy strategic positions, acting as key articulators in international collaborations. A sustained pattern of regional cooperation is observed among Asian and European countries, as well as a consolidated network in the Ibero-American axis, which includes Brazil, Mexico, Portugal, and Argentina. This configuration suggests an active collaborative dynamic driven by common interests in sustainability, innovation in functional foods, and by-product valorization [104].

Figure 8b represents the geographical distribution of scientific productivity, showing a significant concentration of publications in China, followed by Brazil, India, the United States, and Spain. This visualization confirms the existence of regional poles of academic production, mainly in East Asia, Western Europe, and Latin America. The strong participation of Latin American countries suggests a growing orientation toward circular bioeconomy strategies, reflecting the relevance of the topic in regions with high agro-industrial residue generation and considerable potential for their technological utilization [105].

In this regional context, countries such as Chile, Argentina, and Brazil exemplify Latin America’s contribution to this global trend [106,107]. Chile stands out for its progress in valorizing salmon by-products and implementing circular bioeconomy strategies focused on marine resources [108,109]. Argentina, through research led by CONICET, has made notable contributions in the biotechnological production and characterization of bioactive peptides derived from food matrices [110]. Brazil, in turn, leads the region in waste valorization technologies and functional food development, supported by a strong academic and industrial network [111]. These examples illustrate how Latin American countries are advancing sustainable research on protein hydrolysates, combining local raw materials with technological innovation and value-added processes [112].

In this sense, both visualizations reveal an internationalized structure of the field, with a relatively balanced geographical distribution between emerging and established economies. Beyond production volumes, the intensity of scientific collaborations reinforces the interdisciplinary and applied nature of research on protein hydrolysates, consolidating it as a strategic area at the intersection of sustainability, health, and technological development [113]. The collaborative dynamic reflects a globalized field, with China as its epicenter and a growing articulation of Ibero-America, positioning sustainability and the circular bioeconomy as shared interests on an international scale.

### 3.8. Trifactor Analysis (Three-Field Plot)

Figure 9 visually synthesizes the interaction among the most relevant authors, the predominant themes, and the scientific journals in which research on protein hydrolysates and functional properties has been concentrated. This tripartite representation enables the identification of thematic connections that guide research and its dissemination in specialized publication outlets [114].

On the left axis, researchers such as Wubshet SG, Afseth NK, Li Y, Barros L, and Oliveira MBPP are identified for their significant contributions in functional properties, bioactive peptides, and enzymatic hydrolysis. These authors have consolidated their scientific output, aligning it with highly recurrent topics that reflect thematic specialization consistent with the dominant research lines of the field.

The central axis, corresponding to the keywords, shows that the terms most frequently associated with the works of these authors are “functional properties,” “bioactive peptides,” “enzymatic hydrolysis,” and “antioxidant activity,” reinforcing the centrality of these concepts in the scientific discourse of the field. These terms serve as conceptual axes that structure the published research, highlighting a dual focus on the functional validation of the peptides produced and their production via enzymatic technologies [115].

On the right axis, the leading journals of publication are grouped, including Foods, Food Chemistry, Molecules, and LWT—Food Science and Technology, indicating that the most influential contributions have been disseminated through outlets with high impact in the field of food science and biotechnology. This thematic and editorial alignment reinforces the maturity of the field, which is consolidated around an active scientific community unified by recurring topics and a coherent selection of dissemination channels [116].

The figure provides an integrated visualization of how the primary authors have contributed to the most relevant topics and the outlets in which they have disseminated their findings, offering a visual summary of the field’s cognitive and editorial structure. This evidence suggests an apparent convergence between authorial leadership, thematic centrality, and the strategic selection of scientific journals, indicating a well-articulated and expanding field. The three-factor analysis confirms the alignment between leading authors, central topics, and key journals, reflecting the strength and cohesion of the scientific community around protein hydrolysates.

### 3.9. Limitations and Future Work

This study presents limitations inherent to its design. The bibliometric analysis was restricted to the (WoS) database, to articles published in English between 2015 and June 2025. Consequently, relevant works in other languages or indexed in different databases (e.g., Scopus or Dimensions) may not have been captured. Moreover, the absence of motor themes in the thematic map could be related to the threshold parameters applied in Biblioshiny/VOSviewer, which may have limited the inclusion of low-frequency but conceptually relevant keywords. In addition, the 2025 data were only partially indexed at the time of collection, which may slightly underestimate the scientific output for that year.

Despite these limitations, the analysis provides a representative and robust overview of global scientific production on protein hydrolysates and their functional properties. Future research could integrate data from multiple databases to increase coverage, refine threshold parameters to detect emerging themes better, and combine bibliometric approaches with systematic reviews or meta-analyses to deepen conceptual interpretation. Likewise, it would be valuable to explore the link between enzymatic hydrolysis, sustainability metrics, and bioeconomic indicators, promoting the development of predictive models and policies aligned with circular bioeconomy principles.

Looking ahead, several strategic pathways are identified: (i) strengthening the density of basic themes through standardized protocols and functional validations; (ii) transferring sustainability principles into concrete technological processes of biorefinery and functional ingredient development; (iii) combining in silico approaches with experimental confirmations to accelerate the design of bioactive peptides; and (iv) advancing the standardization of evaluation metrics (DH, peptide profiles, bioassays), which will facilitate cross-sectional comparisons and meta-analyses. In addition, future studies should emphasize the industrial and nutritional relevance of protein hydrolysates, particularly their application in functional food formulation, nutraceutical design, and the sustainable recovery of proteins from agro-industrial by-products. These practical implications will strengthen the connection between research outcomes and real-world innovation in the food and biotechnology sectors.

The findings confirm that this is a mature yet expanding field, grounded in the bioactivity of protein hydrolysates and increasingly oriented toward sustainability, methodological innovation, and functional validation. This integration will be key to transforming basic lines into consolidated drivers of scientific research and industrial innovation.

## 4. Conclusions

The bibliometric analysis revealed sustained growth in research on enzymatic protein hydrolysis and its functional properties, with a focus on bioactive peptides, functional properties, and antioxidant activity as central themes, alongside the emergence of approaches linked to sustainability, circular economy, and in silico methodologies. Production was led by Asian and Latin American countries, supported by traditional powers, and disseminated primarily through established journals in the food field, reflecting a mature, collaborative international network. Overall, the results offer a strategic vision of the field by identifying both consolidated areas and those with potential to become drivers of research, providing a valuable basis for guiding future investigations, strengthening innovation, and informing decision-making in food science and technology. From a practical standpoint, fish, dairy, and plant proteins, particularly soy, pea, and rice bran, have emerged as the most investigated sources for producing bioactive peptides with antioxidant and antihypertensive potential. Enzymatic hydrolysis, frequently enhanced by emerging technologies such as ultrasound, high-pressure, or membrane-assisted processes, has shown superior efficiency in improving peptide yield and functional performance. These results reinforce the role of sustainable enzymatic bioprocesses in transforming food by-products into value-added ingredients and linking biotechnological innovation with industrial application.

This trend is especially relevant because it bridges fundamental research with technological innovation, enabling the valorization of underused materials such as fish frames, skins, and bones; dairy whey; and plant residues (soy hulls, okara, rice bran). The integration of enzymatic hydrolysis with assisted methods (ultrasound, high-pressure, enzyme immobilization, and membrane separation) provides a sustainable alternative to conventional processes, reducing waste and energy consumption while maintaining peptide bioactivity. Furthermore, these technological advances promote the development of functional ingredients for foods, nutraceuticals, and biomedical formulations, thereby strengthening the industrial relevance and sustainability of bioactive peptide research. From a broader perspective, the bibliometric trends identified in this study align with the United Nations Sustainable Development Goals, particularly SDG 12 (Responsible Consumption and Production), by promoting sustainable resource use, waste valorization, and circular bioeconomy strategies within the food and biotechnology sectors.

## Figures and Tables

**Figure 1 foods-14-03693-f001:**
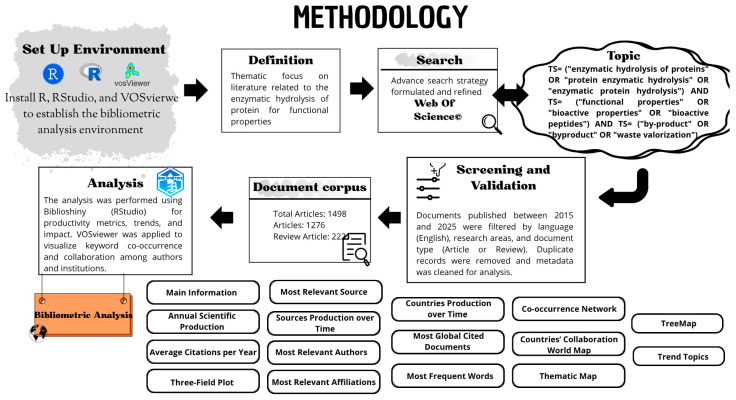
Flowchart of the methodology applied for the bibliometric analysis of scientific literature on the enzymatic hydrolysis and functional properties of proteins.

**Figure 2 foods-14-03693-f002:**
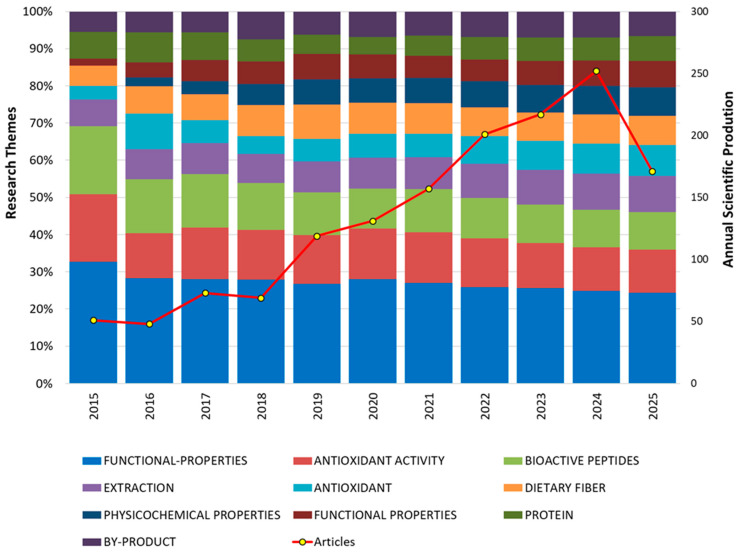
Evolution of scientific publications (2015–2025, collected up to June) on protein hydrolysates from by-products and functional properties. Thematic distribution (%) and the total number of articles are shown by bars and a red line, respectively.

**Figure 3 foods-14-03693-f003:**
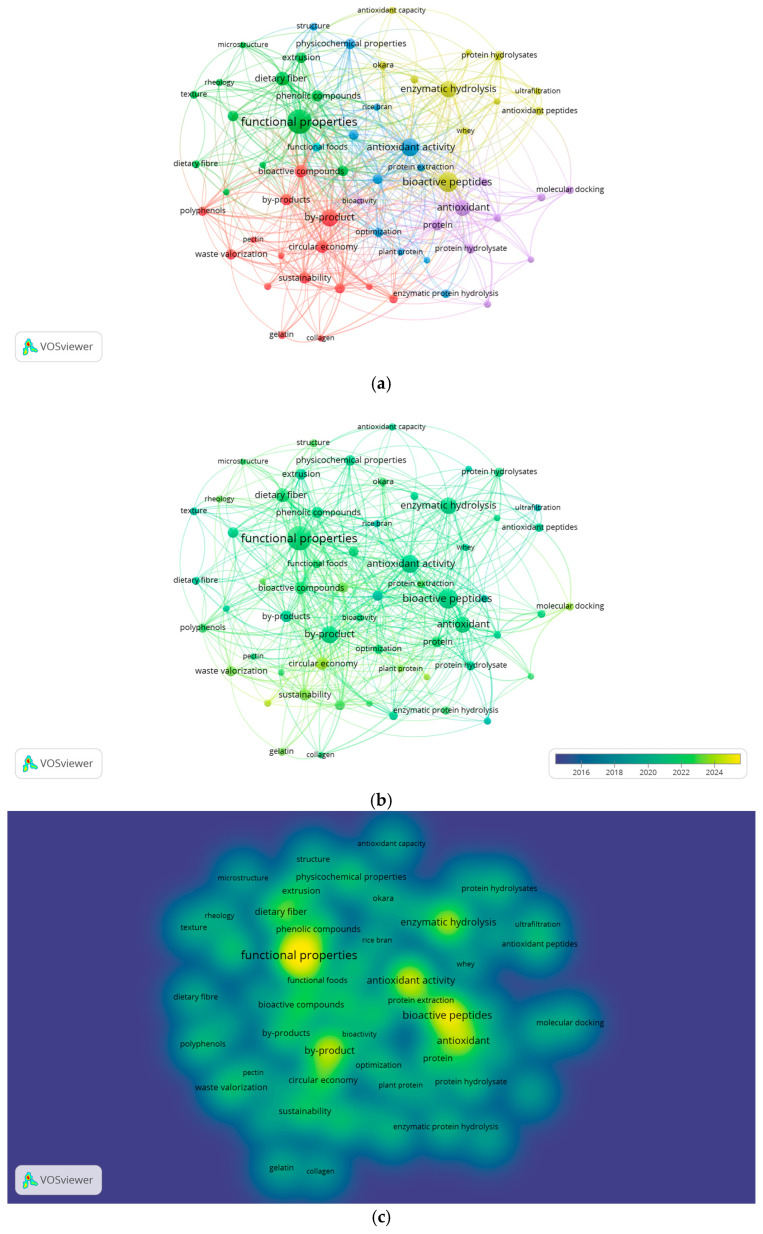
Keyword co-occurrence analysis based on VOSviewer using 1498 indexed articles: (**a**) Network map grouped by thematic clusters (colored nodes); (**b**) Overlay visualization by average publication year (2015–2025, collected up to June); (**c**) Density map of the most frequent terms. The size of each node represents the frequency of occurrence, and the color intensity indicates the density or recency of the term.

**Figure 4 foods-14-03693-f004:**
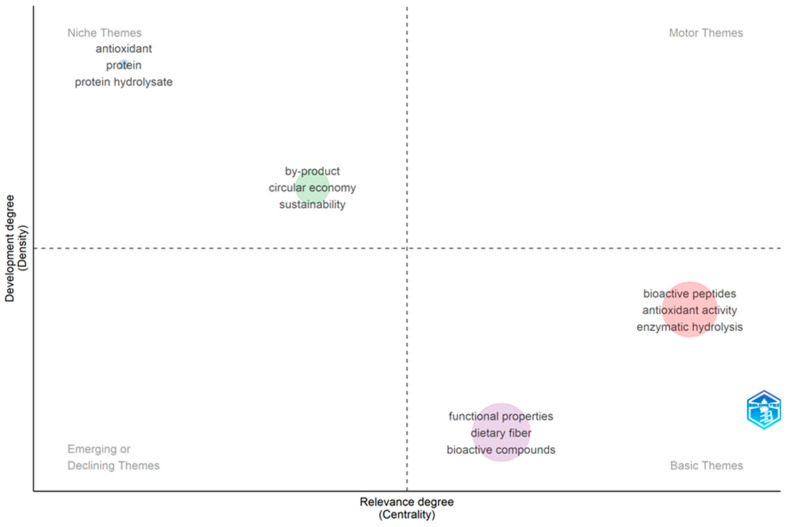
Thematic map of the 50 most frequent keywords based on co-occurrence analysis.

**Figure 5 foods-14-03693-f005:**
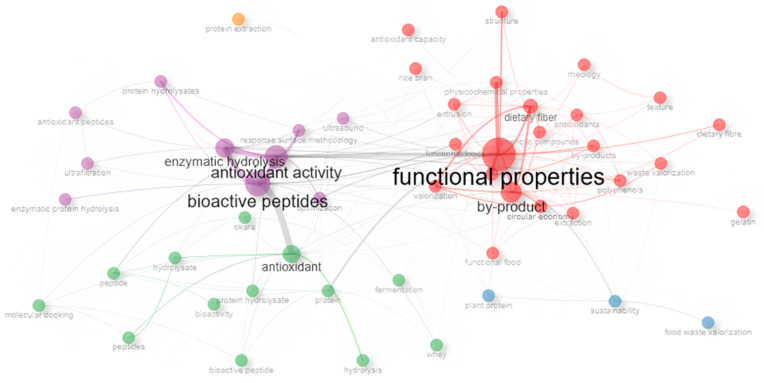
Co-occurrence network of the 50 most frequent author keywords in the field of protein hydrolysates. The figure displays a clustered semantic network generated using the Walktrap algorithm. Node size represents the frequency of each keyword, while the thickness of the edges indicates the strength of co-occurrence relationships. Larger nodes reflect higher keyword frequency, and thicker edges denote stronger conceptual associations between terms.

**Figure 6 foods-14-03693-f006:**
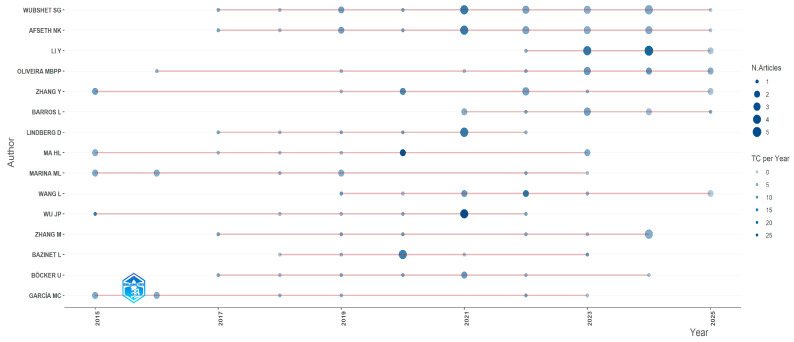
Scientific production of the most prolific authors over time in the field of protein hydrolysates (2015–2025, collected up to June).

**Figure 7 foods-14-03693-f007:**
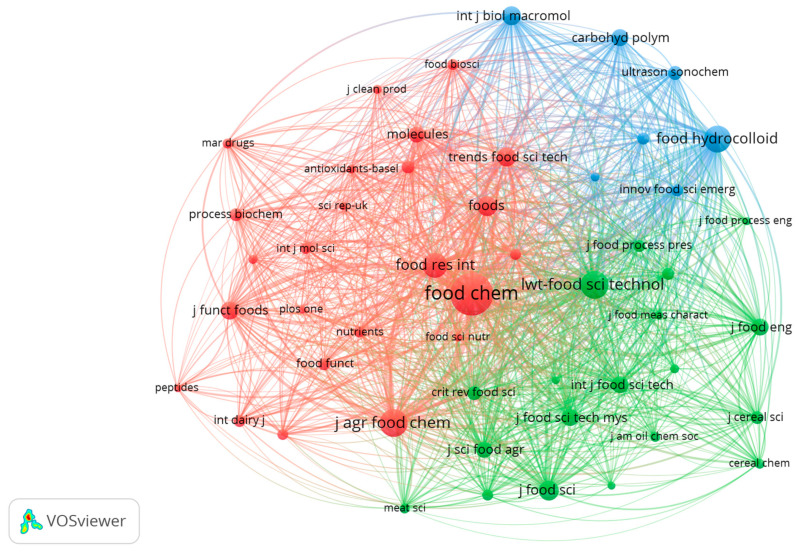
Co-citation network of the 50 most cited journals in the field of protein hydrolysates and functional properties (2015–2025, collected up to June). Node size represents total citations; colors indicate thematic clusters based on citation co-occurrence.

**Figure 8 foods-14-03693-f008:**
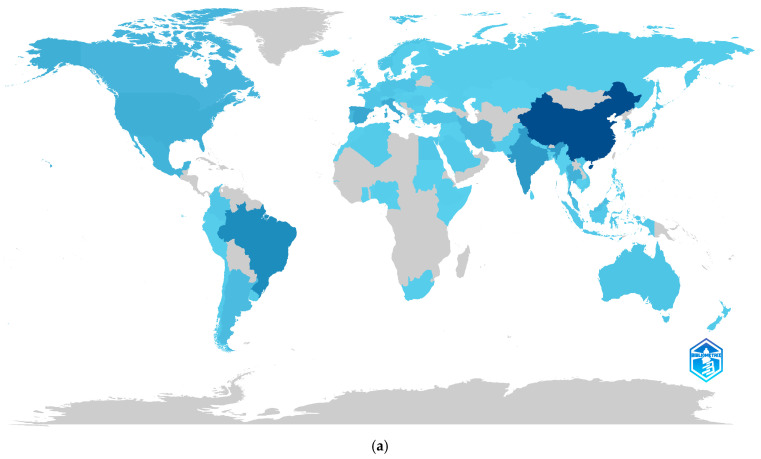
Global scientific collaboration and productivity in the field of protein hydrolysates and functional properties (2015–2025, collected up to June): (**a**) International co-authorship network by country; (**b**) Country-level scientific production based on number of publications.

**Figure 9 foods-14-03693-f009:**
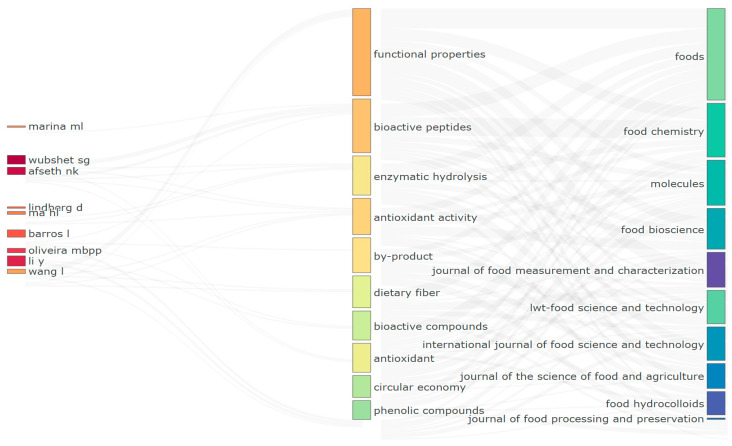
Three-fields plot linking the most relevant authors (left), research themes (center), and publication sources (right) in the scientific production on protein hydrolysates and functional properties (2015–2025, collected up to June).

**Table 1 foods-14-03693-t001:** Overview of the top 10 research lines, leading institutions, countries, journals, and authors according to the number of scientific publications.

**Classification**	**Research lines**	**Number**	**%^a^**
1	Functional-properties	518	34.58
2	Antioxidant activity	250	16.69
3	Bioactive peptides	215	14.35
4	Extraction	207	13.82
5	Antioxidant	177	11.82
6	Dietary fiber	166	11.08
7	Physicochemical properties	165	11.01
8	Functional properties	151	10.08
9	Protein	143	9.55
10	By-product	140	9.35
**Classification**	**Affiliations (Country)**	**Number**	**%^a^**
1	Consejo Superior de Investigaciones Científicas (Spain)	51	3.40
2	Jiangnan University (China)	49	3.27
3	Consejo Nacional de Investigaciones Científicas y Técnicas (Argentina)	45	3.00
4	Indian Council of Agricultural Research (India)	43	2.87
5	Egyptian Knowledge Bank (Egypt)	42	2.80
6	Laval University (Canada)	40	2.67
7	Universidade do Porto (Portugal)	38	2.54
8	Kasetsart University (Thailand)	32	2.14
9	Ministry Of Education (China)	31	2.07
10	Universidade Estadual de Maringa (Brazil)	26	1.74
**Classification**	**Country**	**Number**	**%^a^**
1	China	717	47.86
2	Brazil	354	23.63
3	India	281	18.76
4	Spain	217	14.49
5	Italy	196	13.08
6	USA	173	11.55
7	Thailand	158	10.55
8	Mexico	143	9.55
9	Canada	135	9.01
10	Portugal	134	8.95
**Classification**	**Journals**	**Number**	**%^a^**
1	Foods	110	7.34
2	Lwt-Food science and technology	65	4.34
3	Food chemistry	60	4.01
4	Molecules	44	2.94
5	Food bioscience	42	2.80
6	International journal of food science and technology	42	2.80
7	Journal of the science of food and agriculture	39	2.60
8	Journal of food processing and preservation	37	2.47
9	Food hydrocolloids	36	2.40
10	Journal of food measurement and characterization	35	2.34
**Classification**	**Authors**	**Number**	**%^a^**
1	Wubshet SG	20	1.34
2	Afseth NK	19	1.27
3	Li Y	12	0.80
4	Oliveira MBPP	11	0.73
5	Zhang Y	11	0.73
6	Barros L	9	0.60
7	Lindberg D	9	0.60
8	Ma HL	9	0.60
9	Marina ML	9	0.60
10	Wang L	9	0.60

Note: %^a^ corresponds to the percentage calculated concerning the total number of documents in the corpus (N = 1498) for all classifications, including research lines, affiliations, countries, journals, and authors. The dataset covers articles published between 2015 and 2025, with records indexed until 5 June 2025.

**Table 2 foods-14-03693-t002:** Thematic keyword clusters identified through co-occurrence analysis using VOSviewer.

Cluster	Occurences	Number of Keywords	Keyword in the VOSviewer Network
1 (Red)	403	15	Bioactive compounds, by-product, by-products, circular economy, collagen, extraction, food waste valorization, functional food, functionality, gelatin, pectin, plant proteins, polyphenols, sustainability, waste valorization
2 (Green)	413	12	Antioxidants, chemical composition, dietary fiber, Dietary fibre, extrusion, food waste, functional properties, microstructure, phenolic compounds, rheology, texture, valorization
3 (Blue)	283	11	Antioxidant activity, enzymatic protein hydrolysis, optimization, physicochemical properties, plant protein, protein extraction, response surface, methodology, rice bran, structure, techno-functional properties, ultrasound
4 (Yellow)	323	11	Enzymatic hydrolysis, bioactive peptides, antioxidant capacity, antioxidant peptides, biological activity, fermentation, okara, protein hydrolysates, protein hydrolysis, ultrafiltration, whey
5 (Purple)	227	10	Antioxidant, bioactive peptide, bioactivity, hydrolysate, hydrolysis, molecular docking, peptide, peptides, protein, protein hydrolysate
6 (sky blue)	21	1	Functional foods

**Table 3 foods-14-03693-t003:** Clusters of the most frequent keywords identified by co-occurrence analysis using Biblioshiny.

Cluster	Main Keywords	Number of Keywords	Keywords in the Bibliometrix Network
1	Functional properties	17	Functional properties, by-product, dietary fiber, bioactive compounds, circular economy, phenolic compounds, functional foods, polyphenols, antioxidants, rheology, texture, waste valorization, gelatin, rice bran, structure, dietary fibre, physicochemical properties.
2	Bioactive peptides	13	Bioactive peptides, antioxidant activity, enzymatic hydrolysis, protein hydrolysates, enzymatic protein hydrolysis, antioxidant peptides, biological activity, response surface methodology, optimization, ultrasound, functional food, fermentation, whey
3	Antioxidant	10	Antioxidant, protein, protein hydrolysate, okara, peptides, peptide, hydrolysis, bioactivity, molecular docking, bioactive peptide
4	Sustainability	4	Sustainability, plant protein, food waste valorization, protein extraction
5	Protein extraction	1	Protein extraction

**Table 4 foods-14-03693-t004:** Most frequent keywords by occurrence and total bond strength.

Classification	Keywords	Occurrence	Total Bond Strength
1	functional properties	151	271
2	bioactive peptides	104	201
3	antioxidant activity	84	122
4	by-product	79	240
5	enzymatic hydrolysis	72	98
6	antioxidant	66	156
7	dietary fiber	50	72
8	bioactive compounds	41	27
9	circular economy	41	32
10	phenolic compounds	35	8
11	protein	33	55
12	by-products	32	56
13	sustainability	32	16
14	extrusion	31	3
15	valorization	31	19
16	physicochemical properties	30	2
17	antioxidants	29	10
18	waste valorization	29	3
19	extraction	27	3
20	response surface methodology	27	2

Note: The “Total Bond Strength” corresponds to the sum of co-occurrences of each keyword with all the others within the semantic network, according to the calculations performed in Biblioshiny/VOSviewer.

**Table 5 foods-14-03693-t005:** Highly Cited Scientific Contributions in the Field of Protein Hydrolysates and Functional Properties.

Raking	Title	Source	Year of Publication	Citations	Citations per Year
1	Antioxidant peptides from marine by-products: Isolate, identification and application in food systems [17]	Funct Foods (https://www.sciencedirect.com/journal/journal-of-functional-foods) (accessed on 16 June 2025)	2016	409	40.9
2	Silk sericin: A versatile material for tissue engineering and drug delivery [18]	Biotechnol ADV (https://www.sciencedirect.com/journal/biotechnology-advances) (accessed on 16 June 2025)	2015	348	31.63
3	Cheese Whey: A potential resource to transform into bioprotein, functional/nutritional proteins and bioactive peptides [19]	Biotechnol ADV (https://www.sciencedirect.com/journal/biotechnology-advances) (accessed on 16 June 2025)	2015	277	25.18
4	The seed of industrial hemp (cannabis sativa L.): Nutritional quality and potential functionality for human health and nutrition [20]	Nutrients (https://www.mdpi.com/journal/nutrients) (accessed on 16 June 2025)	2020	271	45.16
5	Fish waste: From problem to valuable resource [21]	Mar Drugs (https://www.mdpi.com/journal/marinedrugs) (accessed on 16 June 2025)	2021	261	52.2
6	Whey as a source of peptides with remarkable biological activities [69]	Food Res Int (https://www.sciencedirect.com/journal/food-research-international) (accessed on 16 June 2025)	2015	241	21.90
7	Applications of hemp in textiles, paper industry, insulation and building materials, horticulture, animal nutrition, food and beverages, nutraceuticals, cosmetics and hygiene, medicine, agrochemistry, energy production and environment [71]	Environ Che Lett (https://link.springer.com/journal/10311) (accessed on 16 June 2025)	2020	219	36.5
8	Isolate and characterization of three antioxidant peptides from protein hydrolysate of bluefin leatherjacket (Navodon septentrionalis) heads [70]	Funct Foods (https://www.sciencedirect.com/journal/journal-of-functional-foods) (accessed on 16 June 2025)	2015	213	19.36
9	The potential of selected Agri-Food loss and waste to contribute to a circular economy: Applications in the food, cosmetic and pharmaceutical industries [80]	Molecules (https://www.mdpi.com/journal/molecules) (accessed on 16 June 2025)	2021	203	40.6
10	Peptides from fish by-product protein hydrolysates and its functional properties [81]	Mar Biotechnol (https://link.springer.com/journal/10126) (accessed on 16 June 2025)	2018	181	22.62

## Data Availability

No new data were created or analyzed in this study.

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
