# Peer review of "Bibliometric Analysis of the Scientific Productivity on Functional Properties and Enzymatic Hydrolysis of Proteins from By-Products"

_foods, 2025, doi:10.3390/foods14213693_

Round 1

Reviewer 1 Report

Comments and Suggestions for Authors

This manuscript is based on the web of science database system to analyze the research field of obtaining protein hydrolysates and their functional properties from agricultural industrial by-products by enzymatic hydrolysis technology. The content of the manuscript has made great progress after modification. In order to further improve its scientific rigor, the following amendments are proposed to the manuscript, and it is recommended to modify the details.

  1. In the introduction part, the necessity of studying the enzymatic hydrolysis technology to obtain protein hydrolysates and their functional properties is not elaborated in depth. It is not deep enough to discuss why special literature statistical analysis should be carried out in this field.
  2. The text description in the chart and text is not clear enough. The description of the chart is simple and not deep enough.
  3. The summary mentions ' 23 most influential participants '. Please supplement the criteria for determining influence.
  4. The paper points out that ' sustainability, circular economy, molecular docking ' and so on are emerging trends. In addition to showing that it was published in a newer year, whether the number or rate of growth of these terms in recent years should be increased to better support the ' emerging ' judgment.

Author Response

Response to Reviewers and Editor:

Manuscript Ref: foods-3948111

The authors would like to thank the editor, and the reviewers for thoroughly assessing the manuscript. We have addressed all comments, and additions are marked in red font in the manuscript version that includes the changes.

Response to Reviewer #1’s Comments

This manuscript is based on the web of science database system to analyze the research field of obtaining protein hydrolysates and their functional properties from agricultural industrial by-products by enzymatic hydrolysis technology. The content of the manuscript has made great progress after modification. In order to further improve its scientific rigor, the following amendments are proposed to the manuscript, and it is recommended to modify the details.

  1. In the introduction part, the necessity of studying the enzymatic hydrolysis technology to obtain protein hydrolysates and their functional properties is not elaborated in depth. It is not deep enough to discuss why special literature statistical analysis should be carried out in this field.

We have delved into the rationale for studying enzymatic hydrolysis technology for obtaining protein hydrolysates and understanding their functional properties. The requested modifications have been incorporated in the revised manuscript at lines 43–46, 50–58, 63–66, 72–74, 81–85, 88–93, 105-108 and 111–113.

  1. The text description in the chart and text is not clear enough. The description of the chart is simple and not deep enough.

We appreciate your comments. Could you please indicate which specific graphs you are referring to in your comment so we can make the corrections appropriately and accurately?

  1. The summary mentions ' 23 most influential participants '. Please supplement the criteria for determining influence.

We have included what was requested in the corrected version of the amunscrio, lines 258-261.

  1. The paper points out that ' sustainability, circular economy, molecular docking ' and so on are emerging trends. In addition to showing that it was published in a newer year, whether the number or rate of growth of these terms in recent years should be increased to better support the ' emerging ' judgment.

We have reviewed this point, and according to the trend analysis, the term “circular economy” shows a recent appearance (2023–2024) with 41 occurrences, while “sustainability” and “molecular docking” record 32 and 17 occurrences, respectively, in the VOSviewer co-occurrence network. These findings indicate a growing scientific focus on sustainable valorization strategies and in silico validation methodologies, suggesting their emergent character within the analyzed field. To clarify this point, a sentence was added in Section 3.2 (Most Relevant Keywords) explicitly noting the recent concentration and frequency of these terms (lines 362-368).

Furthermore, an English-speaking research team member corrected English grammar and sentence structure throughout the manuscript.

Reviewer 2 Report

Comments and Suggestions for Authors

Plaza et al. have aimed to provide a bibliometric analysis of the scientific productivity on functional properties and enzymatic hydrolysis of proteins from by-products by choosing 1498 articles indexed in Web of Science database during 2015-2025. This is an interesting report and should be a good reference database with valuable resource on the reported topic. Therefore, I recommend accepting this article for publication in ‘Foods’ after addressing the following minor comments:

  1. If there is no word limit for ‘abstract’, it is better to include 2-3 more sentences reporting the outcomes/implications of bibliometric analysis.
  2. Two more keywords including “Biblioshiny” and “Valorization” should be included.
  3. Although the research gap and objective are clearly defined, a hypothesis/rationale for why bibliometrics is suitable for this problem should be mentioned.
  4. The version numbers of software such as RStudio, Bibliometrix, VOS viewer should be specified.
  5. Line 242-304: quantify how many occurrences per cluster for more precision.
  6. Line 347-354: the absence of motor themes may be due to threshold parameters, which can be mentioned in limitations.
  7. Line 397-419: As the content is very long, it is better to divide based on thematic blocks for readability.
  8. Line 415-418: The cluster 5 is underdeveloped, but it is better to clarify its scientific significance.
  9. L585-597: Ensure consistency using TC/year or TCpY.
  10. L683-688: Although Latin American context is a valuable contribution, it should be elaborated with examples like Chile, Argentia, Brazil.
  11. L744-747: This limitation statement should be moved to a standalone section ‘Limitations and Future Work’ before conclusion part.
  12. L750-755: this set of future research directions should also include practical implications, that is, industrial or nutritional relevance.
  13. The conclusion should end with a brief statement linking bibliometric trends to global sustainability goals (for example: SDG 12: Responsible Consumption and Production).
  14. All the references should be double-checked for formatting. For example, DOIs, italics etc. Also, the access dates for all online sources should be cited in all figures and table 5.

Author Response

Response to Reviewers and Editor:

Manuscript Ref: foods-3948111

The authors would like to thank the editor, and the reviewers for thoroughly assessing the manuscript. We have addressed all comments, and additions are marked in red font in the manuscript version that includes the changes.

Response to Reviewer #2’s Comments

  1. If there is no word limit for ‘abstract’, it is better to include 2-3 more sentences reporting the outcomes/implications of bibliometric analysis.

We appreciate the reviewer’s valuable suggestion. However, according to the journal’s author guidelines, the abstract has a maximum limit of 200 words. Therefore, we maintained a concise version that summarizes the main objectives, methods, and key findings of the bibliometric analysis while staying within the required length. The primary outcomes and implications are instead detailed in the Discussion and Conclusion sections.

  1. Two more keywords including “Biblioshiny” and “Valorization” should be included.

We have included the keywords “Biblioshiny” and “Valorization” (line 35).

  1. Although the research gap and objective are clearly defined, a hypothesis/rationale for why bibliometrics is suitable for this problem should be mentioned.

We have included a hypothesis/rationale for why bibliometrics is suitable for this problem in the improved version of the document (lines 94-100).

  1. The version numbers of software such as RStudio, Bibliometrix, VOS viewer should be specified.

Poner respuest….. We appreciate the reviewer’s valuable suggestion. The manuscript has been updated to specify the version numbers of all software used. The following versions were added: RStudio (v2025.05.1), R (v4.5.1), Bibliometrix (v5.0.1), and VOSviewer (v1.6.20). These details have been included in the Materials and Methods section (lines 157, 174, and 188).

  1. Line 242-304: quantify how many occurrences per cluster for more precision.

We have quantified the total number of keyword occurrences for each thematic cluster. This information was added to Table 2, providing a clearer quantitative representation of the co-occurrence network and allowing a more precise comparison between clusters. The updated values correspond to the following totals: Red = 403, Green = 413, Blue = 283, Yellow = 323, Purple = 227, and Sky Blue = 21.

  1. Line 347-354: the absence of motor themes may be due to threshold parameters, which can be mentioned in limitations.

We thank the reviewer for this valuable observation. The manuscript has been revised accordingly. A clarification has been added in the Limitations section (lines 780–787), indicating that the identification of thematic clusters and motor themes depended on the threshold parameters established in Biblioshiny/VOSviewer, which may have restricted the inclusion of low-frequency but potentially emerging concepts.

7- Line 397-419: As the content is very long, it is better to divide based on thematic blocks for readability.

We appreciate the reviewer’s helpful suggestion. The section describing the five thematic clusters (Figure 5 and Table 3) has been revised to improve readability. The text was divided into five separate thematic blocks, each corresponding to one cluster: Cluster 1 – Functional Properties (line 443), Cluster 2 – Bioactive Peptides (line 451), Cluster 3 – Antioxidant (line 457), Cluster 4 – Sustainability (line 465), and Cluster 5 – Protein Extraction (line 469). This restructuring enhances clarity and facilitates the interpretation of the thematic segmentation.

  1. Line 415-418: The cluster 5 is underdeveloped, but it is better to clarify its scientific significance.

We have revised the description of Cluster 5 to clarify its scientific relevance. A paragraph was added explaining that, although Cluster 5 (“protein extraction”) shows low connectivity, it represents a specialized methodological area focused on improving protein recovery and purification from agri-food by-products. The revised text highlights its potential contribution to optimizing extraction techniques and linking upstream processing with subsequent hydrolysis and functional validation. This clarification has been incorporated into the manuscript (lines 472–477).

  1. L585-597: Ensure consistency using TC/year or TCpY.

We have corrected the citation metric notation in the text. The term “TCpY” (Total Citations per Year) was adopted for consistency in the corresponding sentence (at line 605).

  1. L683-688: Although Latin American context is a valuable contribution, it should be elaborated with examples like Chile, Argentia, Brazil.

We have revised the paragraph to elaborate on the Latin American context and strengthen its connection with the global discussion. The section now includes examples from Chile, Argentina, and Brazil, which illustrate regional advances in protein hydrolysate research and by-product valorization. Chile was highlighted for its progress in the valorization of salmon by-products and the implementation of circular bioeconomy strategies; Argentina for its contributions through CONICET in the biotechnological production and characterization of bioactive peptides; and Brazil for its leadership in waste valorization and functional food development. These examples were incorporated to create a logical transition toward the subsequent paragraph discussing international collaboration dynamics. The revision appears in (lines 722–731).

  1. L744-747: This limitation statement should be moved to a standalone section ‘Limitations and Future Work’ before conclusion part.

We have created a standalone subsection titled “3.9. Limitations and Future Work”, which now appears before the Conclusions section (lines 779–811). This subsection summarizes the study’s main methodological constraints and outlines perspectives for future research.

  1. L750-755: this set of future research directions should also include practical implications, that is, industrial or nutritional relevance.

In the already constructed subsection “Limitations and Future Work,” the practical implications of future research were incorporated. The text highlights the industrial and nutritional relevance of protein hydrolysates, emphasizing their applications in functional food formulation, nutraceutical development, and sustainable by-product recovery. This addition appears in (lines 802–807).

  1. The conclusion should end with a brief statement linking bibliometric trends to global sustainability goals (for example: SDG 12: Responsible Consumption and Production).

We have expanded the Conclusion section to include a closing statement connecting the bibliometric trends to the United Nations Sustainable Development Goals (SDGs). The revised paragraph now highlights that the field’s research dynamics contribute to SDG 12 (Responsible Consumption and Production) by fostering sustainable resource management, waste valorization, and circular bioeconomy practices in the food and biotechnology sectors. This modification appears in (lines 822–826).

  1. All the references should be double-checked for formatting. For example, DOIs, italics etc. Also, the access dates for all online sources should be cited in all figures and table 5.

We have thoroughly reviewed all the references and verified their formatting, including DOIs, italics, and citation style, ensuring consistency with the journal’s guidelines. All figures and Table 5 were also checked to confirm that the cited online sources include the corresponding access dates.

Furthermore, an English-speaking research team member corrected English grammar and sentence structure throughout the manuscript.

Reviewer 3 Report

Comments and Suggestions for Authors

This article presents a robust and comprehensive bibliometric analysis in the important and rapidly evolving field of enzymatic hydrolysis of proteins from byproducts to yield bioactive peptides. It fills an identified gap by providing an integrated and unified perspective on previously fragmented research areas. The use of advanced tools such as Biblioshiny (RStudio) and VOSviewer to visualize keyword co-occurrence networks and collaboration maps is a strength of the methodology. The overall structure of the article is logical, and the conclusions are well grounded in the presented data.

Section Abstract/IntroductionComment/Area for Improvement Unclear time horizon (2015-2025).Justification The time frame 2015-2025 is given in the introduction and methodology. Since the data was collected up to June 5, 2025, the year 2025 is incomplete, as explained in Section 3.1. This should be more clearly emphasized in the abstract (e.g., "2015-2025, collected up to June") or upon the first use of the range.

Section Methodology (Search Query)Comment/Area for Improvement Simplification of the query in Section 2.Justification In Section 2 (Methodology), the authors state that three thematic sets (TS=...) were combined using the AND operator. However, the full query is technically incorrect in the WoS search language (WoS interprets it as a single string by default). There should be a clearer note in the text explaining that the query was divided into three logical groups connected by AND.

Section Results (Figure 2)Comment/Area for Improvement Incorrect or unclear thematic labels on the cumulative graph.Justification Figure 2 (Cumulative graph) displays the terms "FUNCTIONAL-PROPERTIES" and "FUNCTIONAL PROPERTIES". The authors use the general term "functional properties" in the text. The terms "ANTIOXIDANT ACTIVITY" and "ANTIOXIDANT" also appear. The nomenclature on the graph and in the text should be unified to avoid the impression of category duplication.

Section Results (Figure 3a and 3b)Comment/Area for Improvement Missing Figure 3 in the provided file.Justification Although the review is based on information from the text, the visualization for Figure 3 (keyword clusters and temporal map), to which the text refers, is missing. If possible, Figure 3 should be added for a full verification of the interpretation of clusters and trends.

Section Table 1Comment/Area for Improvement Unclear percentage calculation (%a).Justification The note for Table 1 states that %a for research lines and affiliations was calculated based on the total number of occurrences or articles corresponding to the top ten items in each category. This method is unconventional and can be confusing. Typically, % is calculated relative to the total number of documents in the corpus (1498) or the total number of occurrences for a given category (e.g., total number of keyword occurrences). It would be worthwhile to clarify or recalculate %a for the research lines based on the overall number of articles (N=1498) for clarity.

Section Style and LanguageComment/Area for Improvement Review language:Justification Ensure that section headings, editorial annotations (e.g., Academic Editor: Firstname Last-name, Citation: To be added by editorial staff), and the FOR PEER REVIEW tags are removed in the final version of the article.

Author Response

Response to Reviewers and Editor:

Manuscript Ref: foods-3948111

The authors would like to thank the editor, and the reviewers for thoroughly assessing the manuscript. We have addressed all comments, and additions are marked in red font in the manuscript version that includes the changes.

Response to Reviewer #3’s Comments

This article presents a robust and comprehensive bibliometric analysis in the important and rapidly evolving field of enzymatic hydrolysis of proteins from byproducts to yield bioactive peptides. It fills an identified gap by providing an integrated and unified perspective on previously fragmented research areas. The use of advanced tools such as Biblioshiny (RStudio) and VOSviewer to visualize keyword co-occurrence networks and collaboration maps is a strength of the methodology. The overall structure of the article is logical, and the conclusions are well grounded in the presented data.

Section Abstract/IntroductionComment/Area for Improvement Unclear time horizon (2015-2025).Justification The time frame 2015-2025 is given in the introduction and methodology. Since the data was collected up to June 5, 2025, the year 2025 is incomplete, as explained in Section 3.1. This should be more clearly emphasized in the abstract (e.g., "2015-2025, collected up to June") or upon the first use of the range.

We have more clearly emphasized throughout the manuscript the time the bibliographic data were collected (lines 22, 102, 193, 204, 213, 330, 374, 544, 610, 649, 656, 695, 720, and 756).

Section Methodology (Search Query)Comment/Area for Improvement Simplification of the query in Section 2.Justification In Section 2 (Methodology), the authors state that three thematic sets (TS=...) were combined using the AND operator. However, the full query is technically incorrect in the WoS search language (WoS interprets it as a single string by default). There should be a clearer note in the text explaining that the query was divided into three logical groups connected by AND.

We have revised the search strategy description to clarify that the Web of Science (WoS) query was constructed through three independent thematic topic sets (TS) combined with logical AND operators, ensuring proper Boolean structure.

A clarifying note has been added to Section 2 indicating that each thematic block was entered separately in WoS and combined using AND to obtain the final dataset (lines 126–129).

Section Results (Figure 2)Comment/Area for Improvement Incorrect or unclear thematic labels on the cumulative graph.Justification Figure 2 (Cumulative graph) displays the terms "FUNCTIONAL-PROPERTIES" and "FUNCTIONAL PROPERTIES". The authors use the general term "functional properties" in the text. The terms "ANTIOXIDANT ACTIVITY" and "ANTIOXIDANT" also appear. The nomenclature on the graph and in the text should be unified to avoid the impression of category duplication.

We have reviewed the consistency between the thematic labels presented in Figure 2 and those mentioned in the text. The terms “Functional Properties”, “Functional-Properties”, “Antioxidant Activity”, and “Antioxidant” were maintained as they appear in the original Web of Science database, since their duplication results from differences in keyword indexing. A clarification was added in the text (lines 222–225) to explain this distinction and preserve the fidelity and accuracy of the bibliometric data.

Section Results (Figure 3a and 3b)Comment/Area for Improvement Missing Figure 3 in the provided file.Justification Although the review is based on information from the text, the visualization for Figure 3 (keyword clusters and temporal map), to which the text refers, is missing. If possible, Figure 3 should be added for a full verification of the interpretation of clusters and trends.

We have verified that Figure 3 is included in the manuscript as three complementary panels (Figure 3a, 3b, and 3c) which together present the keyword co-occurrence analysis performed in VOSviewer using the 1498 indexed articles. Specifically, these panels display the network map grouped by thematic clusters, the overlay visualization by average publication year (2015–2025), and the density map of the most frequent terms. The figure and its caption are clearly visible and correctly referenced in the revised version, so the omission may have resulted from a display or download error in the reviewed file.

Section Table 1Comment/Area for Improvement Unclear percentage calculation (%a).Justification The note for Table 1 states that %a for research lines and affiliations was calculated based on the total number of occurrences or articles corresponding to the top ten items in each category. This method is unconventional and can be confusing. Typically, % is calculated relative to the total number of documents in the corpus (1498) or the total number of occurrences for a given category (e.g., total number of keyword occurrences). It would be worthwhile to clarify or recalculate %a for the research lines based on the overall number of articles (N=1498) for clarity.

We have thoroughly reviewed the methodology for the percentages presented in Table 1 was thoroughly reviewed. All %a values were recalculated using the total number of analyzed documents (N = 1498) as the denominator, covering all classifications: research lines, affiliations, countries, journals, and authors. This update ensures methodological consistency and comparability of the results across categories.

Additionally, (lines 234–235) in the manuscript were corrected to update the percentage values corresponding to the lines of research, aligning them with the recalculated data presented in Table 1.

Finally, (lines 291–294) were modified to update the table footnote and reflect these methodological adjustments.

Section Style and LanguageComment/Area for Improvement Review language:Justification Ensure that section headings, editorial annotations (e.g., Academic Editor: Firstname Last-name, Citation: To be added by editorial staff), and the FOR PEER REVIEW tags are removed in the final version of the article.

Thank you for your observation. We have carefully reviewed the manuscript and ensured that all section headings, editorial annotations, and the “FOR PEER REVIEW” tags have been removed in the revised version.

Furthermore, an English-speaking research team member corrected English grammar and sentence structure throughout the manuscript.
